# A High-Power 170 GHz in-Phase Power-Combing Frequency Doubler Based on Schottky Diodes

**DOI:** 10.3390/mi14081530

**Published:** 2023-07-30

**Authors:** Li Wang, Dehai Zhang, Jin Meng, Haotian Zhu

**Affiliations:** 1The CAS Key Laboratory of Microwave Remote Sensing, National Space Science Center, Chinese Academy of Sciences, Beijing 100190, China; wangli223@mails.ucas.ac.cn (L.W.); mengjin@mirslab.cn (J.M.); zhuhaotian@nssc.ac.cn (H.Z.); 2University of Chinese Academy of Sciences, Beijing 100049, China

**Keywords:** doubler, in-phase power combining structure, high power, Schottky diode

## Abstract

In this paper, a high-power 170 GHz frequency doubler based on a Schottky diode is proposed using an in-phase power-combining structure. Unlike a conventional power-combining frequency doubler, the proposed frequency doubler utilizes the combination of a T-junction power divider and two bend waveguides to eliminate the phase difference between the two output ports of the T-junction power divider, so as to achieve in-phase power combining with a concise structure. The frequency doubler was fabricated on a 50 μm thick AlN high-thermal-conductivity substrate to reduce the impact of the thermal effect on the performance. The measured results show that the doubler exhibits a conversion efficiency of 11–31.3% in the 165–180 GHz band under 350–400 mW of input power, and a 118 mW peak output power with a 31.3% efficiency was measured at 174 GHz `when the input power was 376 mW. A good agreement was achieved between the simulation results and the measured performance of the doubler, which proves the effectiveness of the proposed in-phase power-combining structure.

## 1. Introduction

Terahertz (THz) technology has been used in high-speed communication, the remote sensing of the Earth’s atmosphere, safety imaging, and so on due to its special characteristics, such as a high resolution, atmospheric attenuation characteristics, and low quantum energy [1,2,3,4]. However, how to obtain a stable and reliable high-power broadband terahertz source is the primary problem in terahertz technology research [5]. Currently, there are two main approaches to generating a THz signal: photonics and electronics [6,7]. Methods based on photonics have the problems of a low energy conversion efficiency and low output power [8]. Electronic methods mainly include vacuum electronics and solid-state electronics. Based on vacuum electronics, a THz signal can be generated with a pulse power up to the kilowatt level, but their large size and high production and maintenance costs limit their application [9,10]. Solid-state electronics have become mainstream in the generation of terahertz signals due to their small vacuum, low cost, and operation at room temperature [11].

A frequency multiplier, which is a core component of a THz source, will directly affect the performance of the terahertz source. Multipliers can utilize non-liner devices such as high-electron-mobility transistors (HEMT), heterojunction barrier varactors (HBV), and Schottky barrier diodes (SBD) to generate a harmonic signal. Considering that a HBV is a symmetrical device, it only can be used in odd harmonic frequency multipliers [12]. A HEMT is a three-port device with many parasitic parameters, which is not suitable for the higher terahertz frequency band (>1 THz) [13]. As a result, frequency multipliers based on Schottky diodes are preferred in the THz band, benefiting from the advantages of low parasitic parameters, a high cut-off frequency, and a high stability [14,15].

For the further development of a terahertz source, a high-power local oscillator frequency multiplier is essential. Generally, two main approaches are used, a frequency multiplier based on high-bandgap material (such as GaN) Schottky diodes [16,17] or applying power-combining techniques [18,19]. Meanwhile, the multiplier generates heat dissipation during the operation, which will degrade the performance of the multiplier or even disable the diode. Thus, by using high-thermal-conductivity materials as substrates, the effect of the heat dissipation on the performance of the frequency multiplier can be reduced, resulting in an improvement in the output power of the frequency multiplier [20].

The conversion efficiency of a frequency multiplier based on GaN Schottky is limited by its large series resistance. To enhance this efficiency, in this paper, a high-power 170 GHz in-phase power-combining frequency doubler based on GaAs Schottky technology is proposed. An improved T-junction structure is designed to eliminate the phase difference between the output ports of the T-junction. This elimination is realized by adding a bend waveguide after the output ports of the T-junction. Compared to the 3 dB coupling bridge, this structure has the merits of a simplified structure, low processing difficulty, and high-amplitude phase consistency within broadband. The measured results show that the frequency doubler exhibits a conversion efficiency of 11–31.3% in the 165–180 GHz band under 350–400 mW of input power, proving that the doubler with an in-phase power-combining structure is effective.

## 2. Architecture and Design

A reduction in the power-combining loss is the main design issue for a power-combining frequency multiplier. Formula (1) shows the output power of the n-channel power-combining frequency multiplier [21]. P_av_ is the input power of a single branch. For a symmetrical n-branch power synthesis circuit, when the input signals of each branch are consistent in terms of amplitude and phase, the synthesis efficiency is highest, expressed as η_max_. The amplitudes and phases of different branches will be inconsistent due to processing errors, resulting in a decrease in the synthesis efficiency, as shown in Formula (2). M_b_ is the effect factor of the amplitude on the efficiency, with a value of 10^ΔG/10^, where ΔG is the maximum difference in the driving power between the signals. δ_max_ is the effect factor of the phase on the efficiency and the values are in the range of [0,π/2]. Based on the above analysis, in order to obtain the high-power frequency multiplier, it is necessary to select the concise circuit structure to decrease the processing error. Therefore, the modified T-junction structure is adopted in this paper, which not only reduces the processing difficulty in the THz band, but also ensures the consistency of the output port phase of the power divider.
(1)$$P_{out}=\eta_{c}\sum_{k=1}^{n}P_{av,k}$$
(2)$$\eta_{c}\geq \frac{4M_{b}\;\cos^{2}\;\delta_{\max}}{{\left(1+M_{b}\right)}^{2}}\eta_{\max}$$

Figure 1 shows the basic topology of the 170 GHz in-phase power-combining frequency doubler based on a Schottky diode. The direction of the electric field is twisted by the two bend waveguides, eliminating the 180 degree phase difference caused by the two output ports of the T-junction. Thus, the phase consistency of the input signals of the two-branch frequency doubler is realized. The single-branch frequency doubler is designed as a typical Erickson-style structure [22] to form a balanced structure. In this structure, the input signal of the diodes is in the TE10 mode and then transformed into the quasi-TEM mode, and the structure of the frequency doubler is simplified using mode isolation. The single-branch frequency doubler is composed of an input/output-matching circuit and diode model. In the doubler, the input/output-matching circuit, together with the diode model, achieve a high conversion efficiency. Finally, the two-branch doubler output signals are combined by the in-phase power-combining structure to obtain the high output power.

### 2.1. Diode Model and Thermal Characterization

The frequency multiplier uses the non-linear effect of the diodes to generate the required harmonics. Therefore, it is essential to select the suitable diode for the corresponding band. In this paper, the varactor diode produced by ACST with three anodes was chosen. The main parameters of the diode are shown in Table 1.

An essential concern for high-power THz multipliers is heat dissipation, especially at the anode areas of Schottky diodes. Excessive heat dissipation around the Schottky junction due to the high driving power and limited conversion rate can degrade the diode performance or even disable the diode. The effect of temperature on the electrical performance of the diode is shown in Formulas (3)–(5), where k is the Boltzmann constant and χ is the temperature coefficient, which is generally set at 0.3–0.4. V_T_ is the thermal voltage [23]. Formulas (3)–(5) show that temperature causes an increase in the intrinsic parameter Rs and the reverse current Is, leading to a deterioration in the performance of the frequency doubler.
(3)$$V_{T}(T)=\frac{kT}{q}\propto T$$
(4)$$I_{{}_{s}}(T)\propto T^{2}\times \exp(\frac{-\phi_{b}}{V_{T}})$$
(5)$$\begin{array}{cc} R_{s}(T) & =R_{epi}(T)+R_{spreading}(T)+R_{content}+R_{finger}\\ & \approx \chi R_{s}(T)+(1-\chi )R_{s}\\ & \approx \chi R_{s}(T=300K)\times {\left(\frac{T}{300K}\right)}^{0.89}+(1-\chi )R_{s} \end{array}$$

In the terahertz frequency band, the frequency multipliers based on hybrid integration technology usually use a low-relative-permittivity quartz substrate. However, when the input power increases and the thermal effect dominates, the low-thermal-conductivity quartz substrate becomes unsuitable for high-power applications. Therefore, an AlN substrate was chosen for the 170 GHz high-power frequency doubler in this paper, which has a thermal conductivity more than tenfold higher than the quartz substrate. A 3D electromagnetic model of a diode based on the AlN substrate is shown in Figure 2. To more intuitively demonstrate the superior heat dissipation of the AlN substrate compared to the quartz substrate, a thermal simulation was performed using the COMSOL software (COMSOL Multiphysics 5.5’) on the 3D electromagnetic model of the diode shown in Figure 2. The material characteristics in the 3D electromagnetic model of the diode are shown in Table 2. From the thermal simulation results of the 3D electromagnetic model based on the AlN substrate and quartz substrate in Figure 3, it can be seen that, when the dissipation power of a single anode junction of the diode is greater than 15 mW, the junction temperature of the 3D electromagnetic model of the diode based on the AlN substrate is significantly lower than that based on the quartz substrate. When the dissipated power of a single anode is 35 mW, the AlN substrate can reduce the maximum junction temperature by 32 K compared to the quartz substrate. Formula (6) shows the relationship between the dissipated power and the anode junction temperature.
(6)$$T_{anode}-T_{0}=R_{th}(T_{anode})\times P_{dis}$$

### 2.2. Impedance Extraction and Matching Circuit

The efficiency of the frequency multiplier is optimal when the matching circuit and 3D electromagnetic mode of the diode reach a conjugate matching state. The values of the input-matching circuit and the output-matching circuit that need to be conjugated to the 3D electromagnetic model of the diode are called Zin and Zout, as shown in Figure 4. To extract the conjugate value of the embedded impedances of the 3D electromagnetic model of the diode, the in-band (160–180 GHz) embedded impedance optimization method is used, where the embedded impedances Zin is 75 + j∗170 Ω, Zout is 138 − j∗44 Ω.

The input-matching circuit is shown in Figure 5a. For the input-matching circuit, the impedance transformers with narrow side sizes, W2 and W3, are inserted between W1 and W4 to form shunt capacitors in the input-matching circuit [24], where W1 is the narrow side of the waveguide WR-10 and W4 has the same size as the input port of the 3D EM of the diode. By adjusting the lengths of W2 and W3, the input impedance value Zin of the input-matching circuit is shifted in the Smith chart. Finally, the value of Zin that has a subtle error of 75 + j∗170 Ω is obtained. As shown in Figure 5b, the output-matching circuit consists of an output probe and a DC bias filter circuit. The DC bias filter effectively prevents the reflection of the RF signal to the DC port and has little effect on the output impedance value. Thus, the output impedance value of the output-matching circuit can reach the desired value, mainly by adjusting the size of the output probe, and a better output-matching circuit can be obtained. It can be seen from the simulation results of the input- and output-matching circuits in Figure 5c that the impedance values of the input- and output-matching circuits are basically consistent with the required optimum embedding impedance, which proves that the input- and output-matching circuits can achieve conjugate matching with the 3D EM of the diode.

### 2.3. Power Divider/Combiner Circuit

The demand of the power divider in this paper is to divide the input signal into two signals of equal amplitude and phase. In the traditional T-junction structure, there is a 180 degree phase difference between the two output ports of the power divider. Two WR-10 bend waveguides are connected to a T-junction to eliminate the phase difference at the output ports by twisting the direction of the electric field in this paper. In the improved power divider, a reflector-matching structure is adopted, as shown in Figure 6, and the metal-matching block is placed on the inside of the two output branches, playing a role in the circuit impedance matching and effectively extending the bandwidth of the power divider [25]. The simulation results of the return loss of the power divider under different metal block widths, W1, are shown in Figure 7a. At W1 = 0.65 mm, the return loss of the power divider is better than 20 dB in the frequency range of 75–95 GHz, which meets the design requirements. Figure 7b shows the phase simulation results of output ports 2 and 3, demonstrating the feasibility of achieving phase consistency by bending the waveguide to twist the electric field. Considering the effect of machining errors on the performance of the power divider, a tolerance analysis on the size of the metal-matching blocks was performed with a step size of 5 μm. As can be seen from the simulation results in Figure 8, compared to L1, the return loss is more sensitive to W1. With a negative tolerance of W1, the return loss is slightly reduced for bands better than 20 dB.

The power combiner also uses a reflector-matching structure, as shown in Figure 9. In order to expand the bandwidth of the power combiner, a multi-stage waveguide impedance converter structure is adopted. As shown in Figure 9, the combiner has a stepped pattern, which can reduce the effect of discontinuity caused by connecting different sizes of waveguide and achieve bandwidth expansion. The symmetrical structure of the in-phase power-combining frequency doubler determines that the distance L of the power combiner should be the same as that of the power divider, L = 6.17 mm. The phase of input ports 1 and 2 is consistent, as shown in Figure 10a. Figure 10b shows a comparison of the simulation results of the return loss of the power combiner for different-order impedance converters. In the 145–185 GHz frequency range, the return loss of the power combiner based on a two-stage waveguide impedance converter is better than 25 dB, and the bandwidth increases threefold compared to the single-stage one when the return loss is better than 25 dB. Figure 11 shows the results of the tolerance analysis of the metal-matching blocks in the power combiner. According to the results of the tolerance analysis, the performance of the power combiner at a high frequency (>180 GHz) will deteriorate slightly, if the metal-matching block has machining errors.

## 3. Fabrication and Measurements

Figure 12 shows a photograph of the frequency doubler. It can be seem from Figure 12b that the Schottky diodes were flip soldered onto the high-thermal-conductivity AlN substrate. Finally, the frequency doubler was packaged in a 30 mm × 26 mm × 20 mm split metal block, as shown in Figure 12a. To verify the proposed in-phase power-combining doubler, the 170 GHz frequency doubler was measured. The measured platform of the doubler is shown in Figure 13. A ×6 multiplier chain, an adjustable attenuator, and VDI’s PM4 power meter were included in the measured platform. The ×6 multiplier chain could amplify the signal generated by the signal generator up to 28 dBm; thus, a W-band adjustable attenuator with an attenuation factor of 0–30 dB was necessary to adjust the input power to a suitable power level.

A comparison between the measured and simulated conversion loss under 350–400 mW of input power is shown in Figure 14. The measured results show that the conversion loss is less than 10.2 dB (efficiency > 11%) at 165–180 GHz and the best result is 5.28 dB (efficiency = 31.3%) at 174 GHz. Considering the loss of the standard waveguide (estimated at 0.4 dB) and the tapering waveguide, WR5.1-to-WR10 (estimated at 0.4 dB) are not corrected in the measured results. For comparison, the corrected conversion loss is also shown in Figure 14. From Figure 15, it can be seen that the peak output power of the doubler at 174 GHz is 118 mW under 376 mW of input power. To provide an intuitive understanding of the operating state of the frequency doubler, a single-point measurement was performed on the frequency doubler at a different driving power. The single-point measured results are shown in Figure 16. When the input power is higher than 300 mW, the output power of the frequency doubler increases relatively slowly. The reason for this is that the performance of the frequency doubler deteriorates due to the heat dissipation.

Table 3 summarizes the published doubler performance in the similar frequency range. The frequency multiplier based on a GaN Schottky diode can withstand a high driving power. As reported in [16], the frequency doubler can withstand a maximum driving power of 1100 mW. However, due to the high series resistance of the GaN Schottky diode, the conversion efficiency of the frequency doubler is low. The in-phase power-combining frequency doubler proposed in this paper can not only withstand a higher driving power, but also ensure a high conversion efficiency. From the above comparison, it can be concluded that the proposed frequency doubler has a relatively excellent performance.

## 4. Conclusions

A high-power 170 GHz in-phase power-combining frequency doubler based on a Schottky diode was successfully fabricated and studied in this work. The doubler using the proposed divider has the advantage of a simplified structure and reduces the difficulty of the processing in the THz band, so that the influence of the synthesis efficiency caused by error factors can be reduced and enhance the performance of the doubler. The measured results show that the frequency doubler exhibited a peak output power of 118 mW at 174 GHz under 376 mW of input power, which proves the validity of this in-phase power-combining frequency doubler design.

## Figures and Tables

**Figure 1 micromachines-14-01530-f001:**
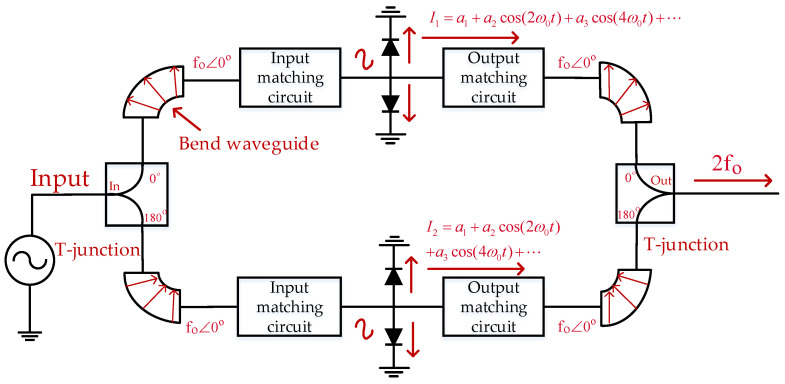
The basic topology of the 170 GHz in-phase power-combining frequency doubler.

**Figure 2 micromachines-14-01530-f002:**
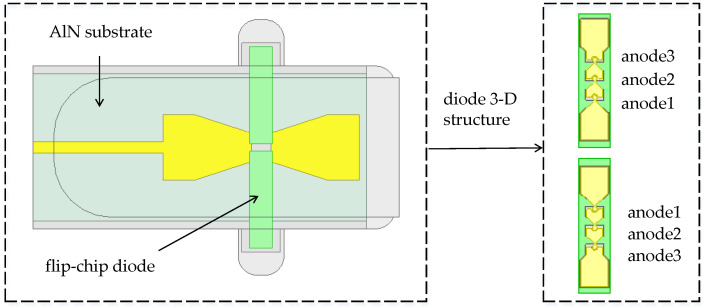
The 3D electromagnetic model of the diode.

**Figure 3 micromachines-14-01530-f003:**
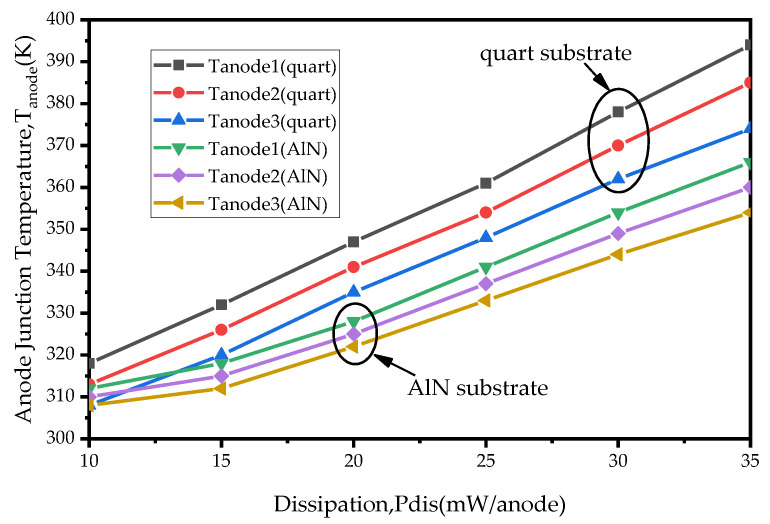
Anode temperature of the 3D electromagnetic model of diode mounted on different substrate.

**Figure 4 micromachines-14-01530-f004:**
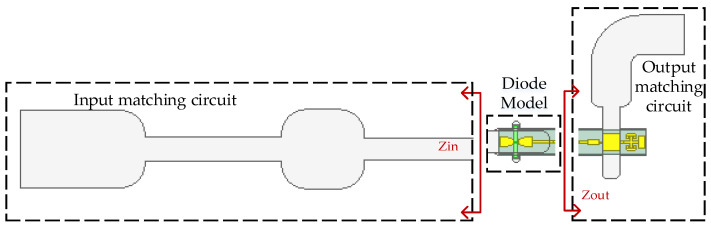
The frequency doubler structure.

**Figure 5 micromachines-14-01530-f005:**
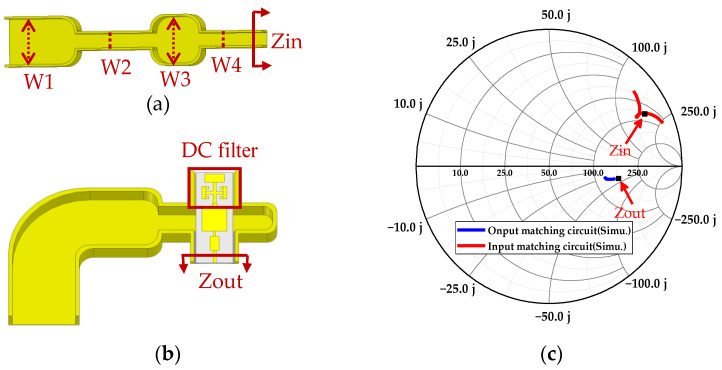
The 3D EM of circuit and simulated results. (**a**) Input-matching circuit (W1 = 1.27 mm, W2 = 0.4 mm, W3 = 1.3 mm, and W4 = 0.36 mm for finial input matching). (**b**) Output-matching circuit. (**c**) the simulated results of circuits.

**Figure 6 micromachines-14-01530-f006:**
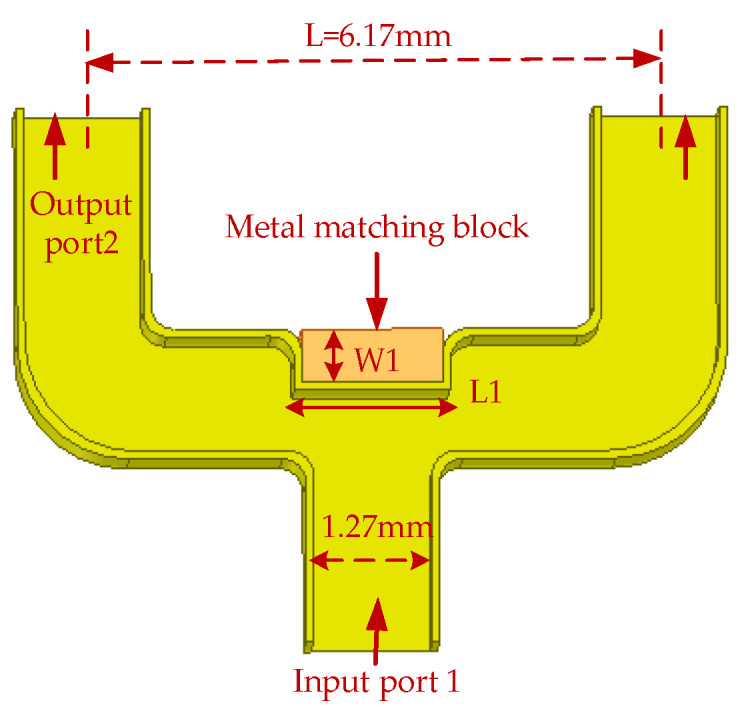
The 3D structure of the power divider (W1 = 0.65 mm, L1 = 0.835 mm).

**Figure 7 micromachines-14-01530-f007:**
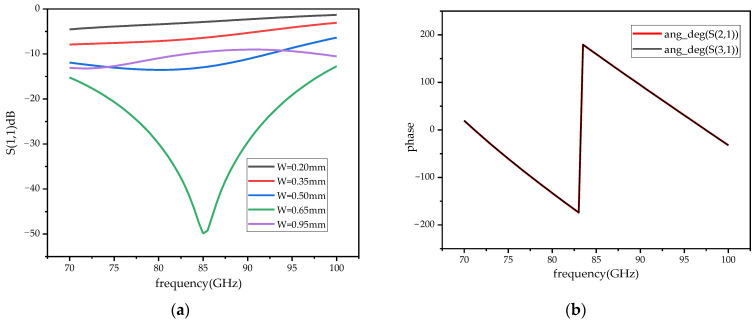
The simulation results of the power divider. (**a**) The return loss simulation results. (**b**) The phase simulation results of the output ports.

**Figure 8 micromachines-14-01530-f008:**
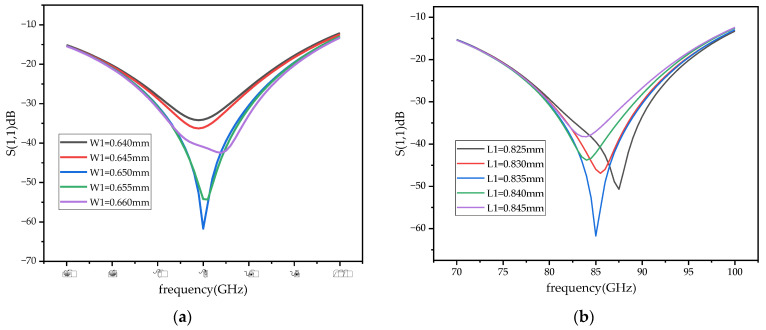
The tolerance analysis the power divider. (**a**) The tolerance analysis of W1 of the metal-matching block. (**b**) The tolerance analysis of L1 of the metal-matching block.

**Figure 9 micromachines-14-01530-f009:**
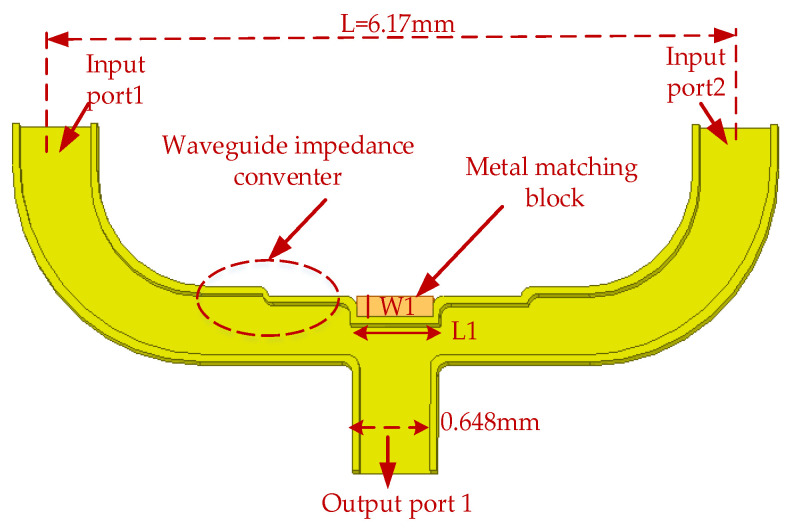
The 3D structure of the power combiner (W1 = 0.18 mm, L1 = 0.404 mm).

**Figure 10 micromachines-14-01530-f010:**
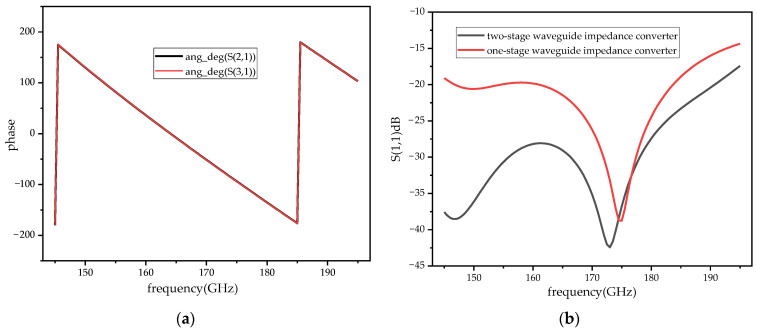
The simulation results of the power combiner. (**a**) The phase simulation results of the input ports. (**b**) The return loss simulation results.

**Figure 11 micromachines-14-01530-f011:**
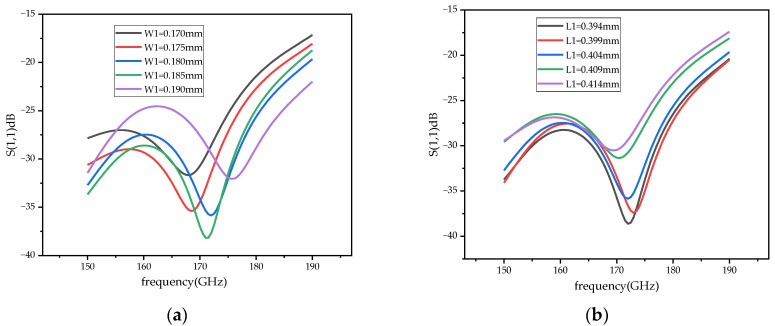
The tolerance analysis the power combiner. (**a**) The tolerance analysis of W1 of the metal-matching block. (**b**) The tolerance analysis of L1 of the metal-matching block.

**Figure 12 micromachines-14-01530-f012:**
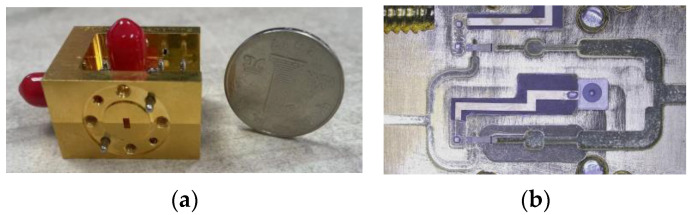
The photograph of the frequency doubler. (**a**) The structure of the frequency doubler. (**b**) The internal structure of the assembled frequency doubler.

**Figure 13 micromachines-14-01530-f013:**
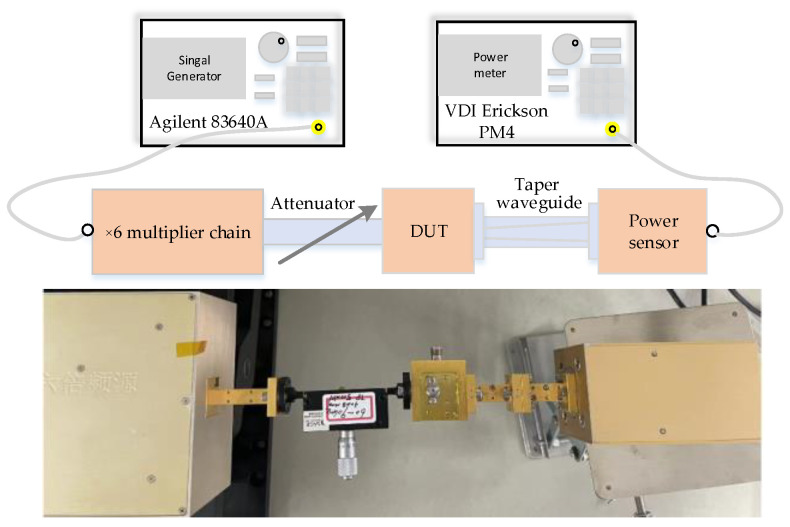
The measured platform of the doubler.

**Figure 14 micromachines-14-01530-f014:**
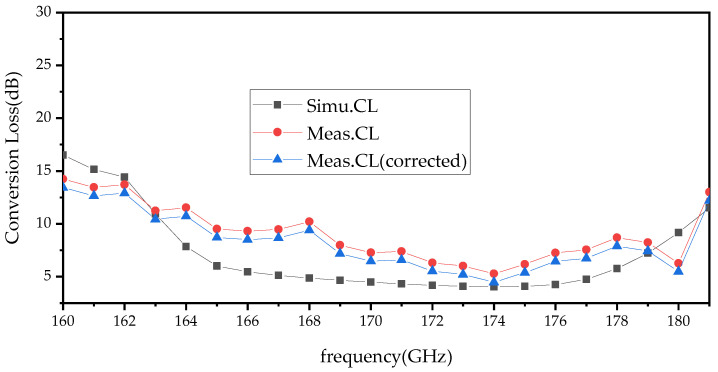
Comparison of simulated and measured conversion loss of the doubler.

**Figure 15 micromachines-14-01530-f015:**
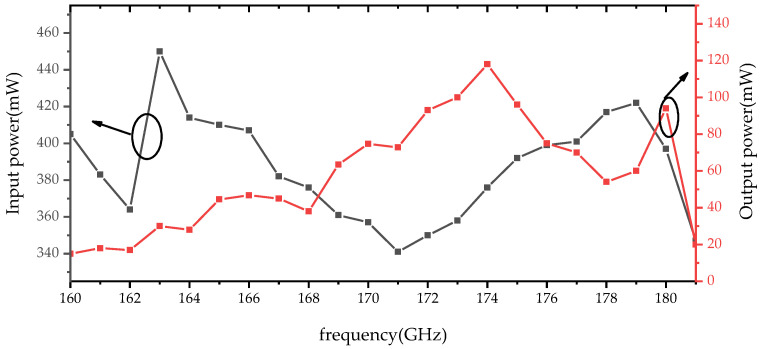
Measured results of the input power (black square line) and output power (red square line) of the doubler.

**Figure 16 micromachines-14-01530-f016:**
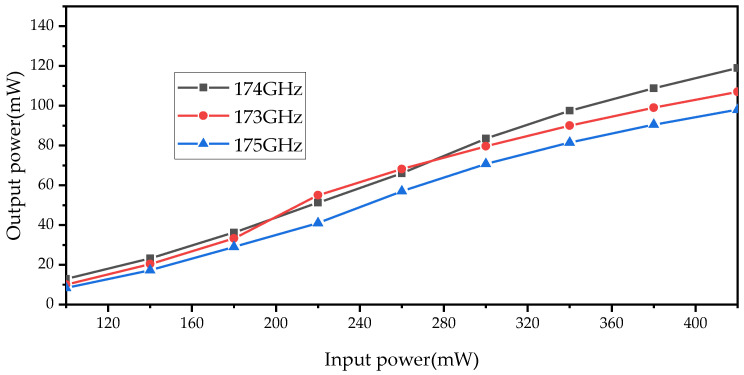
Measured results of the single point of the doubler.

**Table 1 micromachines-14-01530-t001:** The main parameters of the diode.

Is	Reverse saturation current	5.0fA
Rs	Series resistance	4 Ω
N	Emission coefficient	1.2
Cj0	Zero bias junction capacitance	40fF
Vj	Barrier voltage	0.85 V
Bv	Reverse breakdown voltage	13 V
Eg	Band gap width	1.43 eV

**Table 2 micromachines-14-01530-t002:** The material characteristics in the 3D electromagnetic model of the diode.

Material	Thermal Conductivity (W/(m·K))	Bulk Conductivity (S/m)	Relative Permittivity
Metal (Gold)	310	4.1 × 10^7^	1
Insulation layer (SiO_2_)	1.4	0	4.2
Epitaxial layer (GaAs)	51 × (300/T)^1.28^	830	12.9
Buffer layer (GaAs)	51 × (300/T)^1.28^	4.1 × 10^7^	12.9
Substrate (AlN)	160	0	8.8

**Table 3 micromachines-14-01530-t003:** Summary of published doubler performance in the similar frequency range.

	Diode Material	Anodes	Frequency (GHz)	Input Power (mW)	POP * (mW)	PE * (%)
[26]	GaN	8	111–125	286	48.5	17
[27]	GaN	6	117–125	500	15.1	3
[16]	GaN	8	200–220	1100	17.5	1.6
[28]	GaAs	6 × 2	164–172	300–600	59	22
[29]	GaAs	6 × 2	166–179	520–910	204.5	30.2
This work	GaAs	6 × 2	165–180	350–400	118	31.3

* POP: peak output power, PE: peak efficiency.

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
