# Peer review of "A High-Power 170 GHz in-Phase Power-Combing Frequency Doubler Based on Schottky Diodes"

_micromachines, 2023, doi:10.3390/mi14081530_

Round 1
Reviewer 1 Report
1. What is the maximum input power for the present multiplier on AIN substrate leading to the degradation of multiplier performance?
2. There is a typo in Figure 6 and Figure 8 as well ‘Mental Block’.
Reviewer 2 Report
The authors have reported on a a high power 170 GHz frequency doubler based on Schottky diode. The manuscript is interesting, consistent and well written.
Furthermore, the following statements should be reexamined:
1) The power divider is a critical circuit in the design and should be included in the tolerance analysis
2) Figure 13 Maybe it would be better to switch to a tabular form, this is just a suggestion and not a necessity
Minor editing of English language required, please check the expression carefully.
